# Therapeutic Management of Metastatic Clear Cell Renal Cell Carcinoma: A Revolution in Every Decade

**DOI:** 10.3390/cancers14246230

**Published:** 2022-12-17

**Authors:** Mathieu Larroquette, Félix Lefort, Luc Heraudet, Jean-Christophe Bernhard, Alain Ravaud, Charlotte Domblides, Marine Gross-Goupil

**Affiliations:** 1Department of Medical Oncology, University Hospital of Bordeaux, 33000 Bordeaux, France; 2Faculty of Medicine, University of Bordeaux, 33000 Bordeaux, France; 3Department of Urology, University Hospital of Bordeaux, 33000 Bordeaux, France

**Keywords:** checkpoint inhibitor, combination, renal cell carcinoma, strategy, VEGF-TKI

## Abstract

**Simple Summary:**

To summarize the main discoveries made over the past few years to treat metastatic clear cell renal cell carcinoma, including different generations of anti-VEGF TKIs and immune checkpoint inhibitors (ICI), and provide an overview of the future developments to come in this field. We discuss new therapeutic approaches, such as escalation/de-escalation str.

**Abstract:**

Clear cell renal cell carcinoma (RCC) oncogenesis is mainly driven by *VHL* gene inactivation, leading to overexpression of vascular endothelial growth factor (VEGF). The use of tyrosine-kinase inhibitors (TKIs) directed against VEGF and its receptor (VEGFR) revolutionised the management of metastatic renal cancer in the 2000s. The more recent development of next-generation TKIs such as cabozantinib or lenvatinib has made it possible to bypass some of the mechanisms of resistance to first-generation anti-VEGFR TKIs. During the decade 2010–2020, the development of immune checkpoint blockade (ICB) therapies revolutionised the management of many solid cancers, including RCC, in first- and subsequent-line settings. Dual ICB or ICB plus anti-VEGFR TKI combinations are now the standard of care for patients with advanced clear cell RCC. To optimise these combination therapies while preserving patient quality of life, escalation and de-escalation strategies are being evaluated in prospective randomised trials, based on patient selection according to their prognosis risk. Finally, new therapeutic approaches, such as targeting hypoxia-inducible factor (HIF) and the development of innovative treatments using antibody-drug conjugates (ADCs), CAR-T cells, or radiopharmaceuticals, are all potential candidates to improve further patient survival.

## 1. Introduction

For decades, treatment options for metastatic clear cell renal cell carcinoma (mccRCC) have been limited because this chemoresistant cancer was not ideal for early empirical therapeutic approaches. Since the 2000s, a better understanding of oncogenic pathways has led to a paradigm shift in drug development, leading to targeted therapies. mccRCC is mainly driven by *VHL* inactivation causing HIF 1 and 2 overexpression and eventually VEGF release and neoangiogenesis [1]. Other pathways and receptors such as PI3K/AKT/mTor, MET, and AXL, as well as immune escape, have been implicated in tumour growth [2]. In this review, we describe the main treatments and strategies that have emerged as our knowledge has steadily increased, and those that could change clinical approaches to mccRCC treatment in the future. 

## 2. First Revolution: VEGF/VEGFR Inhibitors

The first systemic treatments that were demonstrated to be efficient in mccRCC were the cytokines interleukine-2 (IL-2) and interferon alpha (INF-α), with modest median progression-free survival (PFS) improvement. Although approximately 8% of patients showed complete response (CR) with high doses of IL-2, this benefit came at the price of numerous and severe side effects. In the 2000s, a better understanding of the angiogenic pathway and its implications in kidney cancer led to the development of anti-angiogenic drugs. Bevacizumab in combination with INF-α first showed PFS improvement [3,4], but was soon replaced by the first generation of anti-angiogenic tyrosine kinase inhibitors (TKIs): sunitinib, pazopanib, and sorafenib [5,6,7]. m-Tor inhibitors also demonstrated efficacy in mccRCC during this period [8,9]. 

Since 2012, second-generation anti-angiogenics provided additional options for situations where first-line TKIs failed. First, axitinib showed efficacy in the second-line setting [10]. Multikinase inhibitors allowed these treatments to overcome anti-angiogenic resistance, with cabozantinib targeting VEGFR, and MET and AXL showing OS benefits in the second-line setting in the METEOR phase III trial [11]. Lenvatinib in combination with everolimus was shown to target VEGFR, FGFR, PDGFR-α, KIT, and RET after progression under VEGFR inhibitors, with improved PFS, in a phase II trial [12]. Tivozanib was also designed to selectively inhibit VEGFR1–3 at very low concentrations, showing clinical activity in mccRCC with an interesting safety profile, essentially used in the third or subsequent line of treatment [13,14].

It is important to note that some patients still obtain a benefit from TKI therapy alone during the course of the disease, with the use of successive lines of first and new generations of TKIs providing significant disease control and tumour response. It is crucial to better select these patients, whose disease seems to be driven by neoangiogenesis pathways and who may benefit from long-time TKI treatment, until more effective drugs on other targets become available.

## 3. The Era of Immunotherapy

### 3.1. Immune Checkpoint Blockade (ICB) as Single Agent or in Dual Combination

The immunogenic characteristics of RCC, which are poorly chemosensitive tumours, were first highlighted in the 1990s, with the use of INF-α and IL-2 leading to tumour shrinkage in some metastatic patients [15,16]. However, these therapeutics were responsible for significant high-grade adverse events (AEs) and benefited only a small proportion of patients. 

Targeting the programmed cell death protein 1 (PD-1)/programmed death-ligand 1 (PD-L1) and the cytotoxic T lymphocyte antigen-4 (CTLA-4) axis through the development of ICB therapies has revolutionised the management of several solid tumours over the past decade, such as melanoma and non-small cell lung cancer (NSCLC). PD-L1 expression in RCC, as well as RCC immune sensitivity, were used to evaluate ICB in this cancer subtype [16]. 

In 2015, nivolumab (anti PD-1) became the first US Food and Drug Administration (FDA)-approved ICB for advanced RCC, after showing a benefit in OS over everolimus (median OS, 25.8 vs. 19.7 months; hazard ratio (HR), 0.73; 95% confidence interval (CI), 0.62–0.85) in previously treated RCC in the CheckMate-025 phase III trial [17,18]. Although median PFS was not significantly different between the two groups (4.2 vs. 4.5 months), nivolumab led to long-term responses, with a 36-month PFS probability of 9%, compared with 2% for everolimus. Interestingly, the objective response rate (ORR) was also higher with nivolumab than with everolimus (23% vs. 4%, *p* < 0.0001), with a better tolerance profile (19% vs. 37% grade 3–4 AEs) [17]. 

The efficacy of ICB was then evaluated in the first-line setting, with co-administration of 3 mg/kg nivolumab and 1 mg/kg ipilimumab (anti-CTLA4) for four doses, followed by nivolumab monotherapy, compared to sunitinib (50 mg daily; 4 weeks on, 2 weeks off) for treatment-naïve RCC in the phase III randomised, controlled CheckMate-214 trial [19] (Table 1). The co-primary endpoint results showed significant OS advantage (median OS, 48.1 vs. 26.6 months; HR, 0.65; 95% CI, 0.54–0.78), PFS advantage (median PFS, 11.2 vs. 8.3 months; HR, 0.74; 95% CI, 0.62–0.88), and better ORR (41.9% vs. 26.8%) for combined nivolumab plus ipilimumab compared with sunitinib in the International mRCC Database Consortium (IMDC) intermediate- and poor-risk populations [20]. Nivolumab plus ipilimumab also led to durable responses compared with sunitinib, with 4-year PFS probabilities of 31.0% vs. 17.3% and 32.7% vs. 12.3% in intermediate- and poor-risk disease, respectively [20]. Exploratory analyses in favourable-risk patients showed no OS advantage in the nivolumab plus ipilimumab arm (median OS not reached; HR, 0.93; 95% CI, 0.62–1.40) and found a longer PFS in the sunitinib group (median PFS, 12.4 vs. 28.9 months; HR, 1.84; 95% CI, 1.29–2.62) [20]. These results led to FDA and European Medicines Agency (EMA) approval of nivolumab plus ipilimumab in the first-line setting of intermediate- and poor-risk advanced RCC, making this combination a new standard of care.

### 3.2. TKIs in Combination with ICB 

Despite its good response rates and durable responses, the use of nivolumab as monotherapy led to only 1% CR in the Checkmate-025 trial [17]. The VEGF-driven mechanisms involved in RCC oncogenesis, combined with the evidence that anti-angiogenesis therapy has an immunomodulatory effect, provided a basis for combining ICB with angiogenesis inhibitors in the first-line setting [28]. The associations of VEGF(R) TKI and ICB were evaluated in five phase III randomised, controlled trials (Table 1).

The KEYNOTE-426 trial enrolled mccRCC treatment-naïve patients to receive either pembrolizumab (200 mg/3 weeks) plus axitinib (5 mg twice daily, up to 10 mg twice daily) or sunitinib alone [21,22]. The study met its dual primary endpoints, with updated data showing median PFS and OS improvements of 4.6 and 5.6 months, respectively, in the pembrolizumab plus axitinib arm. In this group, the ORR and CR rate (CRR) were also improved, from 39.6% to 60.4% and 3.5% to 10%, respectively. In subgroup analyses, OS benefit was consistent across all IMDC risk categories (favourable, intermediate, and poor). Combined pembrolizumab plus axitinib showed prolonged efficacy, with a median duration of response of 23.6 vs. 15.3 months and a 42-month OS rate of 57.5% vs. 48.5% compared with sunitinib. Notably, PFS under subsequent line of treatment (PFS2) was longer in the pembrolizumab plus axitinib group (median PFS2, 40.1 vs. 27.7 months; HR, 0.63; 95% CI, 0.53–0.75), in which 47.2% of patients received a subsequent therapy (mainly a VEGF/VEGFR inhibitor), than in the sunitinib group, in which 65.5% of patients received a subsequent line of therapy (mainly an ICB) [23].

In the CHECKMATE-9 ER trial, 651 patients were enrolled to receive either nivolumab (240 mg/2 weeks) plus cabozantinib (40 mg once daily) or sunitinib alone in the first-line setting of advanced ccRCC [24]. The updated results of the study, after a median follow-up of 32.9 months, found a doubled median PFS (16.6 vs. 8.3 m; HR, 0.51; 95% CI, 0.41–0.64), a doubled ORR (55.7% vs. 27.1%), higher CRR (8% vs. 4.6%), and a significant OS advantage (median 37.7 vs. 34.3 months; HR, 0.70; 95% CI, 0.55–0.90) for combined nivolumab plus cabozantinib [24,25].

Lenvatinib (20 mg once daily) combined with pembrolizumab (200 mg/3 weeks) was evaluated, compared to sunitinib, in the first-line setting for mccRCC in the CLEAR trial [26]. After a median follow-up of 26.6 months, median PFS (primary endpoint) was significantly longer in the lenvatinib plus pembrolizumab arm than in the sunitinib arm (23.9 vs. 9.2 months; HR, 0.39; 95% CI, 0.32–0.49). OS was also higher in the lenvatinib plus pembrolizumab group compared to sunitinib, although median OS was not reached in either group (HR, 0.66; 95% CI, 0.49–0.88; *p* = 0.005). Lenvatinib plus pembrolizumab provided higher ORR and CRR than sunitinib (71% vs. 36.1% and 16.1% vs. 4.2%, respectively). Positive results of these four trials with ICB–ICB and ICB–TKI combinations raise the question of the best therapeutic approach for first-line patients. Although cross-trial comparisons should be interpreted with caution, some key differences in these studies are worth noting.

First, the rate of refractory disease (progressive disease as best response) was higher with the use of nivolumab plus ipilimumab (20%) than with a TKI–ICB combination (5.4–10.9%), supporting the use of a TKI–ICB combination for life- or function-threatening disease. Among these combinations, lenvatinib plus pembrolizumab led to the highest CRR (16.1%) and the lowest rate of refractory disease (5.4%), making this combination the best potential therapeutic option if rapid control is needed. By contrast, durable responses obtained with nivolumab plus ipilimumab (“plateau” effects), with PFS of 36% at 2 years and 31% at 4 years, indicate use of this combination in patients with no threatening lesions, to enhance long-term control probability [20].

To date, no predictive biomarkers have been identified to select patients more accurately with regard to those who would obtain greater benefit from dual ICB or ICB–TKI combinations. All 4 studies included patients with PD-L1-positive (≥1%) and negative (<1%) tumours in variable proportions. Subgroup analysis showed a significant survival benefit across all PD-L1 subgroups for all treatment combinations. To note, PFS under nivolumab-ipilimumab combination therapy was more than doubled in the PD-L1 positive subgroup versus the PD-L1 negative subgroup in the Checkmate 214 trial (median PFS, 22.8 months vs. 11 months). This large difference needs to be confirmed in other prospective trials evaluating dual ICB therapy. Concerning histologic subtypes, results on tumours with sarcomatoid features, which are less sensitive to VEGFR TKIs, showed a survival benefit with nivolumab plus ipilimumab over sunitinib in the Checkmate-214 trial [29,30]. Similar results were observed when an ICB was added to a VEGFR TKI with a survival benefit for nivolumab plus cabozantinib over sunitinib in the Checkmate 9ER trial, and for pembrolizumab plus axitinib over sunitinib in the Keynote 426 trial [31,32]. Interestingly, in this trial, CR rates were 11.8% for pembrolizumab plus axitinib and 0% for sunitinib. These results highlight the sensitivity of mccRCC to ICB in this poor prognosis tumour subtype.

Two other negative phase III randomised, controlled trials evaluated angiogenesis inhibitors combined with ICB in the first-line setting: the Javelin Renal 101 trial compared avelumab plus axitinib vs. sunitinib, and the IMmotion151 trial compared atezolizumab plus bevacizumab vs. sunitinib alone [33,34]. Although these studies failed to show an OS advantage for the combination regimen, they provided molecular analyses that contribute to the search for predictive biomarkers.

### 3.3. Therapeutic Response Prediction: Biomarker-Driven Approach

The ability to predict antitumour response remains a challenge in oncology. Combination therapies have proven to be efficient in phase III trials, but we lack reliable tools to determine whether efficacy is driven by either treatment alone or the combination of both for any given patient. The ability to obtain strong negative predictions indicating that a drug will be useless for a specific patient would allow toxicity avoidance and improve patient quality of life and treatment cost-effectiveness.

The IMmotion 150 phase II trial compared atezolizumab with or without bevacizumab or sunitinib in the first-line setting. In an ancillary study, McDermott et al. developed gene expression signatures for T cell effector response (Teff) and angiogenic response (Angio), to assess retrospective tumour responses [35]. In high-Angio patients, median PFS was similar in patients receiving sunitinb (19.5 months) or bevacizumab with atezolizumab (HR, 1.36; *p* = 0.283). In the high-Teff cohort, patients had better median PFS in the combination arm (21.6 months) than in the sunitinib arm (7.8 months) (HR, 0.55; *p* = 0.033). These results have been prospectively confirmed by ancillary studies of the IMmotion151 (atezolizumab plus bevacizumab vs. sunitinib in the first-line setting) and JAVELIN Renal 101 (avelumab plus axitinib vs. sunitinib in the first-line setting) phase III trials [36,37]. Another study predicted anti-angiogenic efficacy using microsRNAs (miRNAs) signatures [38]. Through deep sequencing of 74 mccRCC patients treated with these TKIs, Garcia-Bonas et al. identified 65 miRNAs differentially expressed in tumours progressing under TKI therapy compared with tumours showing at least stable disease. These data were used to build a predictive model for TKI response with a 2 miRNA–based classification showing interesting predictive value on receiver operating characteristic (ROC) curve analysis (AUC = 0.75, 95% CI, 0.64–0.85).

To date, only one randomised trial in mccRCC has proposed patient selection based on molecular features. A 35-gene signature based on genome, transcriptome, and methylome data as well as quantitative reverse transcription-polymerase chain reaction was reported; these data led to the establishment of four molecular subtypes (ccrcc1–4) [39]. The BIONNIKK trial is a multicentre, open-label, randomised multicentre phase II trial [40], with ORR as the primary endpoint. After frozen tissue analysis, patients were classified into groups ccrcc1–4 and randomised accordingly. In total, 202 patients were eligible for the efficacy endpoints and randomised to receive either nivolumab or nivolumab plus ipilimumab in the ccrcc1 and ccrcc4 groups, and nivolumab plus ipilimumab or VEGFR-TKI (sunitinib or pazopanib) in the ccrcc2 and ccrcc3 groups. In the ccrcc1 cohort, ORR was higher with nivolumab and ipilimumab (39%) than with nivolumab alone (29%), with a median PFS of 7.7 and 5.2 months, respectively (HR, 1.27; 95% CI, 0.77–2.11). By contrast, ORR and PFS were similar for nivolumab with or without ipilimumab in the immune-mediated and inflammatory ccrcc4 group. In the ccrcc2 cohort, the ORR was similar for sunitinib or an ICB combination (50 and 51%, respectively) as well as for median PFS (14.4 and 11.1 months, respectively; HR, 0.75; 95% CI, 0.40–1.39), confirming increased sensitivity to anti-angiogenesis in this subgroup. OS data are not yet available.

### 3.4. Tolerance Profiles for Treatment Combinations

Nivolumab plus ipilimumab showed an acceptable toxicity profile in the Checkmate-214 trial, as it was associated with 46% of grade ≥ 3 AEs (mainly immune-related AEs) vs. 63% for sunitinib. FKSI-19 quality of life scores showed a greater mean change from baseline in the nivolumab plus ipilimumab group [19].

The toxicity of VEGFR TKI could theoretically overlap that of ICB when used in combination. The safety results of ICB–TKI combination trials have shown an increase in some grade ≥ 3 AEs under treatment combination, with higher occurrence of diarrhoea (9.1% for pembrolizumab plus axitinib), transaminase elevation (13.3% for pembrolizumab plus axitinib), and hyponatremia (9.4%, for nivolumab plus cabozantinib). Particular attention should be paid to pembrolizumab plus lenvatinib regarding increased grade ≥ 3 hypertension and proteinuria, as observed in the CLEAR trial (27.6% and 7.7%, respectively) [21,24,26]. AE occurrence led to the discontinuation of at least one drug in almost one third of patients, although the effect of discontinuing both drugs was identical between ICB–TKI and solo sunitinib treatment.

## 4. Development of New Therapeutic Strategies

### 4.1. Escalation Strategy

To improve treatment efficacy, therapeutic escalation was studied in the COSMIC-313 trial, in which poor/intermediate-risk patients received nivolumab plus ipilimumab plus cabozantinib vs. nivolumab plus ipilimumab in the first-line setting [27]. Notably, this study was the first in which the control arm used a combination treatment as a comparator. The first results of this study were presented at the European Society for Medical Oncology (ESMO) congress in September 2022, showing an increased median PFS for the triplet association (endpoint not reached vs. 11.3 months; HR, 0.73; 95% CI, 0.57–0.94). Subgroup analysis found that the PFS benefit appeared to be limited to intermediate-risk disease (HR, 0.63; 95% CI, 0.47–0.85), with no PFS improvement in poor-risk disease (HR, 1.04; 95% CI, 0.65–1.69). Although ORR was better with the addition of cabozantinib to nivolumab plus ipilimumab (43% vs. 36%), it was not significantly higher than the 52% ORR observed in the poor/intermediate-risk subgroup of the Checkmate 9ER trial, for the dual association of nivolumab plus cabozantinib. Upcoming OS data with further follow-up results are awaited to clarify the efficacy of this triplet combination for disease management. This intensification strategy was also associated with increased toxicity, with 73% of grade ≥ 3 AEs vs. 41% in the nivolumab plus ipilimumab arm, for a therapeutic benefit that has not yet been well established, especially in poor-risk disease. We believe this triplet combination should not be substituted for the current recommended dual therapy before definitive results from the COSMIC-313 trial, with the aim of preventing increased toxicity.

The ongoing phase III PDIGREE study is also evaluating an escalation strategy, using a sequential approach. The trial includes patients with intermediate/poor-risk disease who received nivolumab plus ipilimumab [41]. After 3 months of treatment, patients with CR undergo nivolumab maintenance, patients with progressive disease (PD) switch to cabozantinib monotherapy, and those with partial response (PR) or stable disease (SD) are randomised between nivolumab monotherapy and intensification with cabozantinib plus nivolumab. The study started in May 2019, with OS as the primary endpoint. The results are anticipated to clarify the role of an intensification strategy for patients with suboptimal response.

### 4.2. De-Escalation Strategy

The optimal duration of ICB treatment, as well as the superiority of dual ICB versus ICB as monotherapy, is poorly understood in advanced RCC, particularly because the efficacy of dual ICB has only been compared to that of TKI treatment. The feasibility of a de-escalation approach, particularly to reduce ICB exposure, was first evaluated in phase II studies.

The OMNIVORE trial was a response-based mono-arm study, in which patients received nivolumab monotherapy [42]. After 6 months of exposure, patients with CR/PR discontinued nivolumab and were observed (arm A), whereas those with SD or PD within the first 6 months received two doses of ipilimumab (arm B). The results showed that 12% of patients (10/83) had a confirmed PR with nivolumab monotherapy. Among patients who discontinued nivolumab, 42% remained nivolumab-free over the first year (arm A) and only 4% of patients converted to a confirmed PR after ipilimumab boosts (arm B).

The mono-arm TITAN-RCC trial evaluated a similar strategy in intermediate/poor-risk disease, with intensification by the addition of 2–4 injections of ipilimumab to nivolumab only in non-responder patients (early PD at week 8 or PD/SD at week 16) under nivolumab monotherapy [43]. In first-line therapy, anti-PD-1 monotherapy led to a 28% ORR, and 40% of patients who presented an early PD with nivolumab monotherapy achieved PR or SD after a nivolumab plus ipilimumab boost.

Finally, in the HCRN GU16-260 phase II study, 117 patients with advanced all-risk RCC received nivolumab as first-line therapy for up to 4 years, achieving an ORR of 29% [44]. A total of 60 patients with PD prior to, or SD after 2 years, were eligible to receive salvage nivolumab plus ipilimumab therapy (four cycles). Among these, 53% received salvage therapy, with 11% responding.

Despite an apparent benefit of ipilimumab boost, with PR obtained in some non-responder patients, these results should be weighed against the 42% ORR observed in the Checkmate 214 trial, with the upfront combination of nivolumab plus ipilimumab. Indeed, early PD could be at risk in some tumour locations. In case of such a de-escalation strategy, selection of patients should be carried out on a case-by-case basis depending on the volume, location, and evolution of the tumour.

The results of the Checkmate 8Y8 trial (active, not recruiting) would help to determine the value of the ICB de-escalation strategy, as this randomised phase III prospective trial assessed the safety and efficacy of nivolumab alone vs. nivolumab plus ipilimumab for first-line treatment of advanced intermediate/poor-risk advanced RCC, with PFS and ORR as primary endpoints (NCT03873402). Ipilimumab boosts were not included in the protocol.

The results of the MOIO phase III trial, a pan-cancer study including metastatic RCC (excluding IMDC favourable-risk patients treated with TKI–ICB), are also anticipated to clarify the non-inferiority of ICB dose spacing (every 3 months) in patients showing CR/PR after 6 months of standard ICB/TKI–ICB therapy (NCT05078047) [45].

Patients with IMDC good-risk disease, or intermediate-risk disease with a single poor prognostic factor, are known to show prolonged survival. A treatment pause in such patients who achieve an objective response under TKI–ICB therapy could significantly improve their quality of life and decrease treatment costs. To evaluate the non-inferiority of such an approach in this low-risk population, the phase III SPICI trial proposes to randomise patients between continued standard therapy and treatment pause for cases showing disease response at 1 year of treatment (NCT05219318).

### 4.3. ICB Re-Challenge Approach in the Second-Line Setting

The survival benefit observed with TKI–ICB in first-line settings suggests a potential synergistic effect of this association. However, its benefit in second-line settings, with ICB re-challenge, remains undetermined. Preliminary retrospective data have highlighted the potential efficacy of ICB rechallenge only in some selected patients [46]. In the phase Ib/II KEYNOTE 146 trial, mccRCC patients received pembrolizumab plus lenvatinib at different line settings (first-line setting, previously treated with ICB, or previously treated with other drugs). ORR at 24 weeks was high in all subgroups and was 55.8% in the 105 patients pre-treated with ICB, with a PD rate of only 3.8% [47]. The FRACTION-RCC phase II trial enrolled patients with mccRCC who were either treatment naïve (track 1) or whose disease previously progressed during or after ICB (track 2). Patients were randomised between nivolumab plus ipilimumab or other ICBs. In total, 46 patients in the track 2 group received nivolumab plus ipilimumab and around half of them had received at least three lines of systemic treatment before enrolment. ORR was 17.4% and PD rate was 30.4%. Even if the response rate was lower than the one observed in Checkmate 214 (39%), the median duration of response of 16.4 months suggests that some patients may still derive clinical benefit from ICB–ICB combination in later treatment lines as well [48].

Two ongoing prospective trials are designed to address this issue. The phase III CONTACT-03 study (NCT04338269), which is aimed at comparing the efficacy of cabozantinib alone vs. cabozantinib plus atezolizumab in patients who experienced disease progression under ICB therapy, is closed for randomisations [49]. The TiNivo2 phase III trial (NCT04987203), which is evaluating tivozanib vs. tivozanib plus nivolumab in the second- or third-line setting, is open for inclusions. [50]

### 4.4. HIF Inhibitors

Improving the survival of patients with metastatic RCC should also involve the development of new classes of drugs, with new mechanisms of action. The majority of ccRCC tumours harbour loss of function of the *VHL* gene, resulting in accumulation of HIF-2α, which upregulates the expression of hypoxia-inducible genes such as *VEGF*, and promotes renal carcinogenesis [51]. The safety and efficacy of belzutifan, a second-generation, orally administered inhibitor of the HIF-2α subunit, was recently evaluated in a dose-escalation/dose-expansion phase I cohort for advanced solid tumours, including 55 patients with heavily pre-treated mccRCC [52]. In mccRCC patients, the ORR reached 25% and median PFS was 14.5 months, with anaemia (27%) and hypoxia (16%) as the most common grade ≥ 3 AEs. Belzutifan was also evaluated in a phase II trial for patients with RCC due to von Hippel–Lindau hereditary disease, resulting in an ORR of 49% (median duration of response not reached), leading to FDA approval of belzutifan for this population in August 2021 [53]. To date, there are three ongoing phase III trials to evaluate the efficacy of belzutifan in advanced RCC, either as monotherapy or in combination with other TKIs/ICBs (NCT04195750, NCT04586231, and NCT04736706).

### 4.5. Immunomodulator Agents

Adaptive T cell response is regulated by the sum of activating and inhibiting signals between antigen-presenting cells and T-lymphocytes, in the immunological synapse. Beyond the CTLA-4/CD80-86 pathway, other inhibitor receptors have been identified as interesting immune checkpoints in renal cell carcinomas [54]. Among them, the T cell immunoreceptor with Ig and ITIM domains (TIGIT) is a member of the Ig superfamily and co-inhibitory receptor that binds CD155 and CD112 and LAG-3, also called CD223. This receptor is upregulated on stimulated T cells to prevent excessive activation and autoimmunity. The use of drugs directed against these new checkpoints has led to encouraging results in solid cancer types, such as nivolumab plus revatlimab (anti LAG-3) in metastatic melanoma, or tiragolumab (anti TIGIT) in non-small cell lung cancer [55,56]. Upcoming results of the randomised phase II trial FRACTION-RCC (NCT02996110), which evaluated nivolumab plus revatlimab against other nivolumab-based combinations in advanced RCC, will provide interesting data on the potential effectiveness of this new antibody [57]. FRACTION-RCC is a multi-arm trial that also aimed to assess the safety and efficacy of linrodostat mesylate, an IDO-1 inhibitor that reduces kynurenine production, in combination with nivolumab.

In contrast to co-inhibitory checkpoints, T lymphocytes express co-stimulatory molecules at their surface, such as OX40, which potentially can be targeted with agonist antibodies. The first results of the randomised phase II study NCT03092856, which evaluated axitinib +/− PFOX (an OX40 agonist antibody) for advanced RCC after prior ICI therapy, showed a non-significant difference between the two arms, but a trend toward better PFS with combination treatment (median PFS, 13.2 vs. 8.5; HR, 0.85; 95% CI, 0.45–1.60) [58]. Several phase I trials are ongoing to assess the safety and preliminary efficacy of OX40 agonists and other co-stimulatory molecules in combination with anti PD(L)-1 antibodies in advanced solid tumours.

Main phase III trials currently enrolling or for which results are pending are summarized in Table 2.

### 4.6. Unsuccessful Leads

#### 4.6.1. IL-Based Therapy

Bempegaldesleukin (NKTR-214) is a pegylated formulation of IL-2 that binds preferentially to the CD122 (IL-2 receptor (IL2R)–βγ) subunits located on CD8^+^ T cells, at the expense of IL2R-α located on Treg cells. Despite promising results in combination with nivolumab in phase I/II trials for metastatic melanoma, the nivolumab/bempegaldesleukin association failed to meet its primary endpoints (ORR and OS) vs. sunitinib or cabozantinib in previously untreated advanced RCC in the PIVOT-09 phase III trial [59]. Given these disappointing results, the bempegaldesleukin plus nivolumab development program was discontinued. The results of a phase III trial evaluating a combination of bempegaldesleukin + nivolumab vs. TKI in the first setting for mccRCC patients are pending (NCT03729245).

#### 4.6.2. Glutaminase Inhibitor

Glutaminase converts glutamine to glutamate, which enters the Krebs cycle and produces energy used by cancer cells. Telaglenastat, an orally administered glutaminase inhibitor, was evaluated in combination with everolimus for pre-treated metastatic RCC in the phase II ENTRATA trial, resulting in a modest 1.9-month increase in median PFS vs. everolimus plus placebo [60]. The primary results of the phase III CANTATA trial, which assessed cabozantinib plus telaglenastat or placebo in pre-treated metastatic RCC, found no significant PFS benefit (primary endpoint) for the telaglenastat association [61].

## 5. Innovative Drugs in Early Development

Innovative drugs with several mechanisms of action are currently under evaluation in early phase clinical trials to treat patients with advanced RCC.

### 5.1. ^177^Lu-Girentuximab

^177^Lu-girentuximab is a radiopharmaceutical monoclonal antibody that targets carbonic anhydrase IX (CAIX), a cell surface protein expressed in ccRCC [62]. ^89^Zr-girentuximab positron emission tomography scanning has been demonstrated to detect ccRCC lesions, providing a rationale for using ^177^Lu-girentuximab as a therapeutic radionucleid [63]. The ongoing single-arm phase II STARLITE-2 trial, with dose escalation followed by an expansion phase, is currently evaluating ^177^Lu-girentuximab plus nivolumab in pre-treated advanced ccRCC (NCT05239533) [64].

### 5.2. SRF388

SRF388 is a fully human IgG1 antibody directed against IL-27, an immunomodulatory cytokine involved in the upregulation of immune checkpoint receptors and downregulation of proinflammatory cytokines [65,66,67]. The primary results of an ongoing phase I/Ib trial have shown a disease control rate of 31% (9/29 patients), including 6 patients with durable disease control at 6 months, in advanced refractory non-small cell lung cancer, hepatocellular carcinoma, and RCC, with an acceptable tolerance profile [68]. The safety and efficacy of SRF388 in combination with pembrolizumab is currently being evaluated in the third part of the trial.

### 5.3. DS-6000a

DS-6000a is an antibody–drug conjugate that binds to cadherin-6 (CDH6), which is overexpressed on the cell surface of RCC and ovarian carcinomas [69]. DS-6000a/CDH6 binding leads to the internalisation of the complex. Then, the cytotoxic drug MAAA-1181a (topoisomerase I inhibitor) is released from DS-6000a to inhibit cell replication and induce cell apoptosis. A phase I clinical trial is currently actively recruiting patients to assess the side effects and the efficacy of DS-6000a in advanced RCC and ovarian carcinomas (NCT04707248) [70].

### 5.4. ALLO-316

ALLO-316 is an allogeneic T lymphocyte engineered to express chimeric antigen receptor (CAR-T) for the CD70 receptor, which is a transmembrane glycoprotein member of the tumour necrosis factor family that is found in various types of cancer cells [71]. ALLO-316 is specifically gene-edited to inactivate the expression of TCRα constant (TRAC) and CD52, with the aim of minimising graft-versus-host disease and conferring resistance to anti-CD52 antibody, which is used in conditioning regimens. Following encouraging preclinical data, the phase I TRAVERSE trial is ongoing to assess the safety and efficacy of ALLO-316 after a lymphodepletion regimen in patients with advanced or metastatic RCC (NCT04696731).

### 5.5. Batiraxcept

AXL expression is associated with anti-angiogenic resistance in RCC. Upon binding with its sole ligand Gas6, AXL promotes angiogenesis and the suppression of innate immune response [72]. Batiraxcept is a high-affinity decoy protein that contains an extracellular region of AXL that binds to Gas6 and then prevents AXL signal activation. The safety and efficacy of batiraxcept, as monotherapy or in combination with cabozantinib or cabozantinib plus nivolumab, are currently being evaluated in a phase Ib/II study in ccRCC [73]. A press release from the manufacturer has announced an ORR of 46% for batiraxcept plus cabozantinib in the first 26 ccRCC metastatic patients enrolled in the study (second- or higher-line therapy) [74].

## 6. Conclusions

Following the advent of the anti-angiogenics and targeted monotherapies, combination treatments (dual immune-oncology or immune-oncology plus TKI) will be likely to remain the first-line standard in mccRCC for years to come. This new paradigm raises several questions for the future. The substantial improvement in OS obtained by combination treatments came with the price of increased toxicity that may not be suitable for real-life patients, who often have greater fragility and more comorbidities. Alternative regimens such as those used in the PRISM trial [75] and de-escalation strategies such as those used in the OMNIVORE or TITAN-RCC trials may address this issue, but more convincing data are needed to alter clinical practice. Another challenging issue is choosing among four first-line options that will never be compared together in a prospective trial. Additionally, we may wonder how to manage treatment sequences after exposure to anti-angiogenics and immunotherapy, particularly when they are used simultaneously, such as in the COSMIC-313 trial. Presently, a new range of drugs is in preclinical or clinical development; some of these appear to be promising, such as the HIF-2α inhibitor belzutifan, which is currently under assessment in phase III trials (MK-6482-011 and MK 6482-005). With this increasingly broad range of treatment options, the next main challenge in oncology will be to deliver the most suitable treatment to each patient. Useful predictive factors are required to enhance drug sensitivity, justify escalation strategies and their increased toxicity risks, and enhance their safety and cost effectiveness. A biomarker-driven approach appears promising, according to the BIONNIKK trial results; however, much time will pass before it is used in routine clinical practice. Eventually, disease management in the metastatic setting may be disrupted by the imminent use of ICB in the adjuvant setting. To date, the positive results of the KEYNOTE 564 trial, which tested pembrolizumab as adjuvant therapy [76] have led to FDA, but not EMA, approval. These favourable results are balanced by the negative results in the adjuvant setting of the IMmotion 010 trial (atezolizumab) [77] and the first disappointing results of the CHECKMATE 914 trial (nivolumab plus ipilimumab), presented at the ESMO congress 2022. Even if there is no current standard, it is interesting to consider the relevance of ICB in the first-line setting, when the patient has relapsed after an adjuvant treatment that used the same mechanism of action.

Despite these issues, the progress accomplished in recent decades and the results of many ongoing trials offer patients and physicians hope for the future.

## Figures and Tables

**Table 1 cancers-14-06230-t001:** Positive results of phase III trials evaluating dual ICB and ICB–VEGFR TKI combination for first-line treatment of advanced RCC.

Study	Treatment Arms	Patients	IMDC	OS	PFS	ORR	PD as Best Response (Refractory)	Median Time to Response	Patients Who Received a Subsequent Line of Treatment
Checkmate 214 [19,20]	Nivolumab + ipilimumab vs. sunitinib	1096	Intermediate and poor	median OS: 48.1 m vs. 28.6 m, HR 0.65, 95 CI [0.54–0.78]	median PFS: 11.2 m vs. 8.3 m, HR 0.74, 95 CI [0.62–0.88]	42% vs. 27%	20% vs. 17%	2.8 m vs. 3 m	39% vs. 54%
Keynote 426 [21,22,23]	Pembrolizumab + axitinib vs. sunitinib	861	All	median OS: 45.7 m vs. 40.1 m, HR 0.73, 95 CI [0.60–0.88]	median PFS: 15.7 vs. 11.1 m, HR 0.68 95 CI [0.58–0.80]	60% vs. 40%	10.9% vs. 17%	2.8 m vs. 2.9 m	47% vs. 66%
Checkmate 9ER [24,25]	Nivolumab + cabozantinib vs. sunitinib	651	All	median OS: 37.7 vs. 34.3, HR 0.70, 95 CI [0.55–0.90]	median PFS: 16.6 vs. 8.3 m, HR 0.56, 95 CI [0.46–0.68]	56% vs. 27%	5.6% vs. 13.7%	2.8 m vs. 4.2 m	25% vs. 40%
CLEAR [26]	Pembrolizumab + lenvatinib vs. sunitinib	712	All	median OS: NR vs. NR, HR 0.66, 95 CI [0.49 to 0.88]	median PFS: 23.9 m vs. 9.2 m (HR 0.39, 95 CI [0.32 to 0.49]	71% vs. 36%	5.4% vs. 14%	1.9 m vs. 1.9 m	55% vs. 71%
COSMIC-313 [27]	Nivolumab + ipilimumab + cabozantinib vs. nivolumab + ipilimumab	855	Intermediate and poor	Immature	median PFS: NR vs. 11.3 m, HR 0.73, 95 CI [0.57 to 0.94]	43% vs. 36%	8% vs. 20%	NR	NR

ICB: immune checkpoint blockade, VEGFR: vascular endothelial growth factor receptor, RCC: renal cell carcinoma, IMDC: International Metastatic RCC Database Consortium, OS: overall survival, PFS: progression-free survival, ORR: objective response rate, PD: progressive disease, HR: hazard ratio, 95 CI: 95% confidence interval, m: months, NR: not yet reported.

**Table 2 cancers-14-06230-t002:** Main phase III trials in mccRCC currently enrolling or for which results are pending.

Study Name	Main Characteristics	Population	Experimental Arm	Comparator Arm	Primary Endpoint	Recruitment Status	Study Number
	**New molecules**						
MK-6482-005	Post-anti- PD(L)1 + post-TKI	736	Belzutifan	Everolimus	PFS, OS	Active, not recruiting	NCT04195750
MK-6482-011	Second line (post-ICB)	708	Belzutifan + lenvatinib	Cabozantinib	PFS	Recruiting	NCT04586231
MK6482-012	First line	1431	Belzutifan + pembrolizumab + lenvatinib and pembrolizumab + quavonlimab + lenvatinib	Pembrolizumab + lenvatinib	PFS, OS	Recruiting	NCT04736706
RENAVIV	First/second line (post-immunotherapy)	413	Pazopanib plus abexinostat	Pazopanib + placebo	PFS	Recruiting	NCT03592472
	**Escalation strategy**						
PDIGREE	First line (int/poor IMDC) According to response after 4 cycles nivolumab + ipilimumab	1046	Non-CR/Non-PD cabozantinib + nivolumab CR: Nivolumab PD: Cabozantinib	Non-CR/Non-PD CR: Nivolumab PD: Cabozantinib	OS	Recruiting	NCT03793166
PROBE	First line	364	Nephrectomy if non-progression at week 10 to 14	Standard treatment	OS	Recruiting	NCT04510597
	**De-escalation strategy**						
Checkmate 8Y8	First line (int/poor IMDC)	473	Nivolumab	Nivolumab + ipilimumab	PFS	Active, not recruiting	NCT03873402
SPICI	First line (fav/int with one IMDC fav criteria only) With OR at 12 Months with PD1/ICB + TKI VEGFR	372	Treatment Pause	Treatment continuation	PFR	Not yet recruiting	NCT05219318
	**Rechallenge strategy**						
CONTACT-03	Post-anti PD(L)1	500	Cabozantinib + atezolizumab	Cabozatinib	PFS, OS	Active, not recruiting	NCT04338269
TINIVO-2	Second/third line, post-ICB	326	Tivozanib + nivolumab	Tivozanib	PFS	Recruiting	NCT04987203

PFS: progression-free survival, int: intermediate, IMDC: International Metastatic renal cell carcinoma Database Consortium, CR: complete response, PD: progressive disease, OS: overall survival, OR: overall response, ICB: immune checkpoint blockade, TKI: tyrosine kinase inhibitor, PFR: progression-free rate.

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
