# Peer review of "Therapeutic Management of Metastatic Clear Cell Renal Cell Carcinoma: A Revolution in Every Decade"

_cancers, 2022, doi:10.3390/cancers14246230_

Round 1
Reviewer 1 Report
Excellent review article of RCC current and future therapies. I have the following comments:
- Table 2: citations are incorrect, please fix
- Table 2: COSMIC 313, best response as progressive disease was reported in the study, please correct
- Page 5: “Concerning histologic subtypes, results on tumours with sarcomatoid features, which are less sensitive to VEGFR TKIs, showed a survival benefit with nivolumab plus ipilimumab over sunitinib in the Checkmate-214 trial, but also with nivolumab plus cabozantinib over sunitinib in the Keynote-426 trial, highlighting the sensitivity of mccRCC to ICB in this poor prognosis tumour subtype”. Nivolumab plus cabozantinib was studied in Checkmate 9ER, please fix. Also please discuss the sarcomatoid population data from KEYNOTE 426 using pembrolizumab plus axitinib https://ascopubs.org/doi/10.1200/JCO.2019.37.15_suppl.4500
- Please change the COMSIC 313 reference (39) to the correct one from the ESMO annual meeting website https://oncologypro.esmo.org/meeting-resources/esmo-congress/phase-iii-study-of-cabozantinib-c-in-combination-with-nivolumab-n-and-ipilimumab-i-in-previously-untreated-advanced-renal-cell-carcinoma-arc.
- ICB rechallenge in the second line setting: Please discuss KEYNOTE 146 data of pembrolizumab plus Lenvatinib in RCC patients who received prior ICB and also FRACTION RCC ipi/nivo post prior ICB exposure (PMID: 34143969, PMID: 36328377)
- Line 369: “Main phase III trials currently enrolling or for witch results are pending are summarized in table 2” Please correct witch to which.
- Table 2: COSMIC 313 and PIVOT 09 studies were reported at ESMO 2022. Please correct and discuss in relevant parts of the manuscript
Author Response
Dear Reviewer
Thank you for your careful reading and wise suggestions
Please find bellow our answers:
Table 2: citations are incorrect, please fix
A : we are not sure we understand: if you meant the reference about the table in the text or the title of the table we fixed it, or do you want us to add citations in the table when available (like article or abstract about study designs ?).
Table 2: COSMIC 313, best response as progressive disease was reported in the study, please correct
A : done
- Page 5: “Concerning histologic subtypes, results on tumours with sarcomatoid features, which are less sensitive to VEGFR TKIs, showed a survival benefit with nivolumab plus ipilimumab over sunitinib in the Checkmate-214 trial, but also with nivolumab plus cabozantinib over sunitinib in the Keynote-426 trial, highlighting the sensitivity of mccRCC to ICB in this poor prognosis tumour subtype”. Nivolumab plus cabozantinib was studied in Checkmate 9ER, please fix. Also please discuss the sarcomatoid population data from KEYNOTE 426 using pembrolizumab plus axitinib
A : Corrections were made and we mentioned the trial about sarcomatoid features in renal cancers from the Keynote 426 trial:
“Concerning histologic subtypes, results on tumours with sarcomatoid features, which are less sensitive to VEGFR TKIs, showed a survival benefit with nivolumab plus ipilimumab over sunitinib in the Checkmate-214 trial [28,29]. Similar results were observed when an ICB was added to a VEGFR TKI with a survival benefit for nivolumab plus cabozantinib over sunitinib in the Checkmate 9ER trial and for pembrolizumab plus axitinib over sunitinib in the Keynote 426 trial [30,31]. Interestingly, in this trial, CR rates were 11.8% for pembrolizumab plus axitinib and 0% for sunitinib. These results highlight the sensitivity of mccRCC to ICB in this poor prognosis tumour subtype.”
Please change the COMSIC 313 reference (39) to the correct one from the ESMO annual meeting
A : Done
ICB rechallenge in the second line setting: Please discuss KEYNOTE 146 data of pembrolizumab plus Lenvatinib in RCC patients who received prior ICB and also FRACTION RCC ipi/nivo post prior ICB exposure (PMID: 34143969, PMID: 36328377)
A : We propose:
In the phase 1b/2 KEYNOTE 146 trial, mccRCC patients received pembrolizumab plus lenvatinib at different line settings (first line setting, previously treated with ICB, or previously treated with other drugs). ORR at 24 weeks was high in all subgroups and of 55.8% in the 105 patients pre-treated with ICB, with a PD rate of only 3.8%. [47] The FRACTION-RCC phase 2 trial, enrolled patients with mccRCC who were either treatment naïve (track 1) or whose disease previously progressed during or after ICB (track 2). Patients were randomised between nivolumab plus ipilimumab or other ICBs. Fourty-six patients in the track 2 group received nivolumab plus ipilimumab and around half of them had received at least three lines of systemic treatment before enrolment. ORR was 17.4% and PD rate was 30.4%. Even if the response rate was lower than the one observed in Checkmate 214 (39%), the median duration of response of 16.4 months suggest that some patients may still derive clinical benefit from ICB-ICB combination in later treatment lines as well. [48]
Line 369: “Main phase III trials currently enrolling or for witch results are pending are summarized in table 2” Please correct witch to which
A : done
Table 2: COSMIC 313 and PIVOT 09 studies were reported at ESMO 2022. Please correct and discuss in relevant parts of the manuscript
A : done : these trials were removed from the table and discussed elsewhere although we could discuss to keep COSMIC 313 since OS data are still pending ?
Thank for your help, time and consideration and if you have any further suggestion please let us know

Reviewer 2 Report
This manuscript by Larroquette and colleagues is a comprehensive review of the progress made to treat metastatic RCC over the past few years. It covers all of the major therapeutic approaches from cytokines to antiangiogenic drugs to TKIs, then transitions into the current era of immunotherapy. They cover the recent trials evaluating combination therapies which have shown great promise, and reference the new therapeutic strategies such as escalation and de-escalation approaches and ICB re-challenge in the second line setting. Finally the new HIF2a inhibitor, immunomodulator agents and several new innovative drugs are described, and in order to be completely comprehensive, they have even reported on unsuccessful leads. This review is a very up-to-date overview of the topic and provides the reader with a wealth of information in one review.
There are only a few minor points to address:
1) Title - does not really describe the content of the article. The word “therapy” needs to be included, I believe. “Therapeutic management of metastatic ccRCC-a revolution in every decade” could be one suggested title.
2) It might be useful to add a few sentences in the immunomodulator agents section to describe what LAG-3 and TIGIT do immunologically for the reader without this background knowledge.
3) Lines 369 and 370-2 spelling errors
4) Table 2-could a column be added to indicate whether a trial is still in enrollment or closed?
5) Tables-I assume the tables will be typeset differently. Right now the titles are inconsistently bold or not, with different sized text in Table 1 vs Table 2.
Author Response
Dear reviewer,
thank you very much for your time and consideration as well as for your suggestions
Here are our answers to your questions/remarks :
1) Title - does not really describe the content of the article. The word “therapy” needs to be included, I believe. “Therapeutic management of metastatic ccRCC-a revolution in every decade” could be one suggested title.
A: done
2) It might be useful to add a few sentences in the immunomodulator agents section to describe what LAG-3 and TIGIT do immunologically for the reader without this background knowledge.
A: we propose :
Adaptive T cell response is regulated by the sum of activating and inhibiting signals between antigen-presenting cells and T-lymphocytes, in the immunological synapse. Beyond the CTLA-4/CD80-86 pathway, other inhibitor receptors have been identified as interesting immune checkpoints in renal cell carcinomas[54]. Among them, the T-cell immunoreceptor with Ig and ITIM domains (TIGIT) is a member of the Ig superfamily and co-inhibitory receptor that binds CD155 and CD112 and LAG-3, also called CD223. This receptor is upregulated on stimulated T-cells to prevent excessive activation and autoimmunity.
3) Lines 369 and 370-2 spelling errors
A: corrected
4) Table 2-could a column be added to indicate whether a trial is still in enrollment or closed?
A : done
5) Tables-I assume the tables will be typeset differently. Right now the titles are inconsistently bold or not, with different sized text in Table 1 vs Table 2.
A : tables have been harmonized, typesets will be corrected with the editor
Thank you again for your helpful suggestions and if you have any further remarks, please let us know

Reviewer 3 Report
Congratulations to the authors for this excellent manuscript. I send you some suggestions in order to improve the quality of the paper:
1. In point 1 "First revolution: VEGF/VEGFR inhibitors", (see line 53), authors says "for the first time, targeted approaches allowed a benefit in overall survival (OS), in this disease". But really, sunitinib, sorafenib and pazopanib could not demonstrated a benefit in OS (yes in PFS). Please, check it.
2. In the next paragraph (line 56), authors write about "options for situations where first-line TKIs failed". In the second-line setting, axitinib, tivozanib, cabozantinib and lenvatinib plus everolimus are named. But really, tivozanib was used in the third/fourth-line setting (see TIVO-3 trial, reference 11). So I suggest to clarify it.
3. In Table 1, there is a erratum. In COSMIC 313, authors say "Nivolulmab + ...". Please, write down "Nivolumab + ...".
4.It is striking that adjuvant setting occupies a large part of the conclusions. But the adjuvant therapy is not named previously in the manuscript. I suggest writing previously about adjuvant therapy or changing the conclusions.
Author Response
Dear reveiwer, thank you for your remarks that helps us to improve our review
Please find attached the corrected manuscript and the answers to your remarks
Thanks again !
- In point 1 "First revolution: VEGF/VEGFR inhibitors", (see line 53), authors says "for the first time, targeted approaches allowed a benefit in overall survival (OS), in this disease". But really, sunitinib, sorafenib and pazopanib could not demonstrated a benefit in OS (yes in PFS). Please, check it.
For the sake of clarity, the sentence has been deleted.
2. In the next paragraph (line 56), authors write about "options for situations where first-line TKIs failed". In the second-line setting, axitinib, tivozanib, cabozantinib and lenvatinib plus everolimus are named. But really, tivozanib was used in the third/fourth-line setting (see TIVO-3 trial, reference 11). So I suggest to clarify it.
The paragraph has been edited to clarify the use of tivozanib in third or subsequent line of treatment.
3. In Table 1, there is a erratum. In COSMIC 313, authors say "Nivolulmab + ...". Please, write down "Nivolumab + ...".
Done
4.It is striking that adjuvant setting occupies a large part of the conclusions. But the adjuvant therapy is not named previously in the manuscript. I suggest writing previously about adjuvant therapy or changing the conclusions."
Given that our manuscript deals with metastatic RCC, we agree that the adjuvant situation should not occupy so much space in the conclusion. However, we feel it is important to mention the potential place of immunotherapy in the adjuvant setting, as this will possibly lead to a change in practice soon. For this reason, we propose to address this topic in a more summary way in the conclusion, as an opening remark (see conclusion paragraph in the manuscript).
